# Factors associated with self-reported diagnosed asthma in urban and rural Malawi: Observations from a population-based study of non-communicable diseases

**Abena S. Amoah**[1,2¤]*, **Estelle McLean**[1,2], **Alison J. Price**[1,2], **Alemayehu Amberbir**[3], **Amelia C. Crampin**[1,2,4]

1 Department of Population Health, Faculty of Epidemiology and Population Health, London School of Hygiene and Tropical Medicine, London, United Kingdom, 2 Malawi Epidemiology and Intervention Research Unit, Chilumba, Malawi, 3 University of Global Health Equity, Kigali, Rwanda, 4 School of Health and Wellbeing, University of Glasgow, Glasgow, United Kingdom

¤ Current address: Leiden University Center for Infectious Diseases, Leiden University Medical Center, Leiden, The Netherlands
* a.s.amoah@lumc.nl

**Data Availability Statement:** All data used in these analyses have been anonymised and made

## Abstract

The growing burden of asthma in low- and middle-income countries has been linked to urbanisation and lifestyle changes. However, this burden has not been well characterised in adults. Therefore, we investigated the prevalence of self-reported diagnosed asthma and associated factors in urban and rural adults in Malawi, Southern Africa. Within a cross-sectional population-based survey to determine the burden and risk factors for non-communicable diseases (NCDs) in the city of Lilongwe and rural Karonga district, we collected information on self-reported previously diagnosed asthma and asthma-related symptoms using an interviewer-led questionnaire. Other data collected included: demographic characteristics, socioeconomic status indicators, NCD comorbidities, environmental exposures, and anthropometric measurements. We used multivariable logistic regression models to explore factors associated with self-reported asthma adjusting for variables associated with the outcome in univariable analysis. Findings were corrected for multiple comparisons using the Bonferroni method. We analysed data from 30,483 adult participants (54.6% urban,45.4% rural and 61.9% female). A prior asthma diagnosis was reported in 5.1% of urban and 4.5% of rural participants. In urban females, being obese (>30 kg/m$^2$) compared to normal weight (18.5–24.9 kg/m$^2$) was associated with greater odds of asthma (OR = 1.59, 95% CI [1.26–2.01], p<0.001), after adjusting for confounders. We observed associations between previously diagnosed heart disease and asthma in female participants which remained significant in rural females after Bonferroni correction (OR = 2.30,95%CI [1.32–4.02], p = 0.003). Among rural males, current smokers had reduced odds of diagnosed asthma (OR = 0.46,95%CI [0.27–0.79], p = 0.004) compared to those who had never smoked. In Malawi the prevalence of self-reported diagnosed asthma was greatest in females and urban dwellers. Notably, our findings indicate relationships between excess body weight as well as comorbidities and diagnosed asthma in females. Future

available for download via the Zenodo Repository (https://zenodo.org/records/12097720).

**Funding:** Funding for this study was from the Wellcome Trust (098610/Z/12/Z and 098610/B/12/A). The funder played no role in the study design, data collection and analysis, decision to publish, or preparation of the manuscript.

**Competing interests:** The authors have declared that no competing interests exist.

investigations using longitudinally collected data and clinical measurements of asthma are needed to better understand these associations.

## Introduction

Asthma is one of the most common chronic conditions [1] affecting an estimated 262 million people worldwide and in 2019, resulted in 455,000 premature deaths [2]. Most asthma-related deaths occur in low- and middle-income countries (LMICs) where health-care resources are constrained, and the condition remains under-diagnosed and under-treated [2].

Asthma is made up of multiple phenotypes and represents complex interactions between genetic and environmental determinants [3]. Factors associated with urbanisation such as greater exposure to environmental pollutants [4], changes in diet [5], a greater body mass index [6, 7] and indoor allergen exposure [8], have all been linked to the rising prevalence in asthma worldwide. In children, the global burden of asthma and associated risk factors have been relatively well-characterised [9–11]. Much less is known about asthma in adults, particularly in LMICs. From observations made primarily in high income countries (HICs), adult-onset asthma is largely non-allergic [12], more common in females [13], associated with an accelerated decline in lung function [12] and linked to obesity [14–16]. The few studies conducted on asthma in LMICs do indicate marked urban-rural variations in asthma burden [17–21]. Initiatives such as the Burden of Obstructive Lung Disease (BOLD) study which collected country-specific data to characterise lung disease in adults worldwide [22, 23] have been addressing the evidence gap between HICs and LMICs. However, diagnostic challenges exist and in many LMICs where the availability of diagnostic tools for asthma are limited, diagnosis and management of the condition rely largely on clinical observations by health-care professionals and reported symptoms thus resulting in under-diagnosis and under-treatment [24].

In Malawi, a low-income country in sub-Saharan Africa, asthma is commonly diagnosed in primary health-care settings [25]. Most published studies from Malawi have been hospital-based and focused on the diagnosis and management of multiple respiratory conditions including asthma [26–28]. Recently, several population-based studies conducted in the Southern region of Malawi investigated lung function, respiratory symptoms, and the burden of noncommunicable lung disease [29–32]. Although not focused on asthma, some of these studies established associations between biomass smoke exposure, smoking, sex and respiratory symptoms in adults in Southern Malawi [29, 30].

The aim of our current study was to investigate the prevalence of self-reported diagnosed asthma in adults. Additionally, we sought to explore factors associated with asthma, including demographic characteristics, anthropometric measurements, socioeconomic status indicators, NCD comorbidities and environmental smoke exposures.

## Methods

### Study design

Between May 2013 and April 2017, the Malawi Epidemiology and Intervention Research Unit (previously known as Karonga Prevention Study) conducted a large population-based cross-sectional investigation of the burden and risk factors for non-communicable diseases (NCDs) in urban and rural adults in Malawi. The design, population descriptions and detailed methodology for this study have been described elsewhere [33, 34]. Published findings from this investigation observed notable differences in the prevalence of NCD outcomes such as

hypertension in urban versus rural area as well as in males versus females [34]. Within this large study, information was also collected on self-reported diagnosed asthma, asthma symptoms and factors potentially associated with asthma.

## Study area and population

For our investigation, the rural study area was within the established Karonga Health and Demographic Surveillance Site (Karonga HDSS) [35] where approximately 50,000 individuals reside in a geographical area of 135km$^2$ in the southern part of Karonga District in the Northern Region of Malawi and are under continuous demographic surveillance. The Karonga HDSS population is predominantly rural with an economy based on subsistence farming, fishing and petty trading [35]. Our urban study area was in Lilongwe, the capital city of Malawi, which is in the Central Region of the country. The study area in Lilongwe was an enumerated zone in a high-density residential area named Area 25 which has an estimated population of 107,316 individuals [36] of mixed socioeconomic status [33]. Enumeration involved collecting data on household membership, location details, and information on individuals ≥18 years [33]. Data collection for our study was between 16 May 2013 and 25 April 2017 in both study areas.

## Data collection

**Study questionnaire.** Questions from the International Union against Tuberculosis and Lung Disease (IUALTD) bronchial symptoms questionnaire [37] were administered to participants along with questions on demographic parameters, socioeconomic indicators, medical history as well as lifestyle factors. These measurements have been used elsewhere in Africa [38].

The specific questions related to asthma and its symptoms were:

1. *Have you ever been diagnosed with asthma, asthmatic bronchitis or allergic bronchitis by a doctor or other health professional*? *If yes, what age were you when first diagnosed*?

2. *Are you on regular medication for asthma*? *If yes, name of medication* ____________

3. *Have you had wheezing or whistling in your chest at any time in the past year*?

4. *In the last year, have you ever had an attack of wheezing or whistling that has made you feel short of breath*?

The study questionnaire was developed in English, translated into Chitumbuka (for the rural area) and Chichewa (for the urban area). It was then independently back translated and tested to ensure understanding and acceptability before implementation. Questionnaires were administered by trained interviewers using electronic tablet computers.

Questions on potential risk factors for asthma such as age, area of residence (urban or rural), indicators of socio-economic position including education level, occupation, household income and ownership of certain possessions, physical activity levels, NCD comorbidities and environmental exposures such as cigarette smoke, firewood smoke exposure and water source were included.

For these analyses, we categorised education based on Malawian educational levels ranging from no formal education, primary school (Standard 1–5 and Standard 6–8), secondary school and tertiary education. Physical activity was based on minutes spent on physical activity according to the World Health Organisation (WHO)'s guidelines [39] and three categories were generated (low, moderate and high physical activity). We generated a wealth score based on possessions (electricity, lamp, mobile phone etc.) and an income score based on reported

income. The NCD comorbidities of interest were previously diagnosed diabetes mellitus, heart disease, high blood pressure and stroke as reported by study participants. For smoke exposures, 'smoking status' was based on individual smoking history while 'smoker in household' gauged second-hand smoke exposure at home. We combined information on firewood use for cooking and whether cooking areas were ventilated or non-ventilated to generate a firewood exposure variable.

**Anthropometric measurements.** For each study participant, we determined weight and height measurements using a calibrated scale and portable stadiometer respectively. Body Mass Index (BMI) was calculated by dividing weight (kg) by height (m) squared and was categorised according to the WHO's BMI classification for adults: underweight (less than 18.5kg/$m^2$), normal weight (18.5–24.9kg/$m^2$), overweight (25.0–29.9kg/$m^2$) and obese (>30.0kg/$m^2$) [40].

## Data management and quality assurance

Data was predominantly captured electronically via protected tablet computers using forms pre-programmed in Open Data Kit (https://getodk.org/) which were uploaded each day to a database housed in secure central server at each research site. Data quality was assured through rigorous training of Interviewers and the use of an interviewer's guide to ensure uniformity of data collection. Pre-programmed data checks were performed in the MEIRU data offices to identify any discrepancies which were addressed immediately. Uploaded data were also regularly extracted and checked for scientific validity. Data management processes were strictly controlled through adherence to set protocols and a central data dictionary [41].

## Informed consent and ethical approval

We provided a full explanation of the study objectives and procedures to potential participants prior to seeking individual written consent. Ethical approval for the study was granted by the Malawi National Health Sciences Research Committee (protocol number #1072) and the London School of Hygiene & Tropical Medicine Ethics Committee (protocol number #6303).

## Statistical analysis

Our primary outcome was self-reported diagnosed asthma. A secondary outcome was reported wheeze in the past 12 months. We also considered other asthma-related outcomes for descriptive analysis. We investigated differences in baseline characteristics stratified by sex and urban-rural area using Pearson's χ2 tests (for categorical variables) and Mann-Whitney U tests (for continuous variables). In preliminary analysis, we evaluated the extent to which associations between our covariates and outcome of asthma varied by sex and urban-rural area. To this end, we generated models fitted with an interaction term between sex and each covariate as well as separate models with interaction terms for urban-rural area. Our exploration showed evidence that the association between some sociodemographic indicators (for example, age-group, BMI, education, and work status) varied by sex and urban-rural area (p<0.05). Therefore, our logistic regression models were generated stratified by sex and urban-rural area. In univariable logistic regression analysis, associations between factors of interest and study outcomes (asthma and wheeze in the past 12 months) were examined. A two-sided p-value threshold of <0.100 was used to determine factors for inclusion in multivariable analysis. Final models were adjusted for all factors found to be significant in univariable analysis. To account for the effect of multiple testing, we applied a Bonferroni correction where alpha of 5% was divided by the number variables in the final logistic regression model [42]. Therefore, a corrected alpha of 0.4% (p = 0.004) was considered as the threshold for statistically significance

for our multivariable regression analyses. All analyses were conducted using STATA version 17 software (StataCorp, College Station, Texas).

## Results

### Characteristics of the study population

A total of 30,483 participants (54.6% urban, 45.4% rural and 61.9% female) were included in the analyses. Table 1 shows characteristics of study participants stratified by sex and urban-rural area. According to WHO BMI classifications, 18.0% of urban females were obese (30.0 kg/m$^2$) compared to 7.7% % of rural females. By contrast, 3.9% of urban males were classified as obese and only 1.0% of rural males. Most individuals in all groups reported a physical activity rating categorized as 'high' according to the WHO physical activity standards.

Compared to rural participants, urban males and females had significantly higher values when it came to markers of socio-economic position (median household income, median household possessions and proportion who completed tertiary education or were self-employed / employed).

With regards to NCD comorbidities, diagnosed high blood pressure was significantly greater in urban females (13.0%) compared to rural females (8.2%). The same pattern was observed for urban versus rural males. Overall, the proportion of participants reporting NCD comorbidities was relatively low (<3%).

Over 35% of urban participants of both sexes reported having piped water as a main source of household water compared to under 10% of rural participants. Current smoking in our study population was low overall but higher in males (11.2%) compared to females (0.2%). More rural females (15.5%) reported being exposed to household second-hand smoke compared to urban females (10.5%). When it came to firewood exposure, more urban females (11.4%) reported firewood smoke exposure in a non-ventilated cooking area compared to rural females (0.6%). The same pattern was observed for urban males (10.1%) compared to rural males (0.4%) for this factor.

### Prevalence of asthma and related symptoms

Diagnosed asthma was reported in 5.1% of urban and 4.5% of rural participants. As shown in Table 2, a significantly higher proportion of urban females (5.6%) reported diagnosed asthma compared to rural females (4.5%) (p = 0.001) but no difference was observed for males (p = 0.871). Median age of asthma diagnosis was higher in rural females (17.0 years, IQR [6.5–28.0]) compared to urban females (14.0 years, IQR [5.0–23.0] and much lower in both male groups. Only 2.1% of participants reported regular use of asthma medication (for example, aminophylline, salbutamol, prednisone, or inhalers). As shown in Fig 1, 71.9% (1067/1484) of those with diagnosed asthma did not report having symptoms of 'wheeze in the past 12 months' or 'wheeze in the past 12 months plus shortness of breath'.

Poor overlap was also seen with the outcome 'diagnosed asthma with medication use' where 59.8% (374/625) did not report having 'wheeze in the past 12 months' or 'wheeze in the past 12 months plus shortness of breath (S1 Fig).

### Factors associated with a self-reported asthma diagnosis

S1 Table shows the results of univariable logistic regression analysis of potential factors associated with a self-reported diagnosed asthma stratified by sex and urban-rural area. The factors found to be associated with asthma (p<0.100) were age-group, BMI, physical activity rating, household income, household asset index, education, work status, diagnosed heart disease,

**Table 1. Characteristics of study population stratified by sex and urban-rural area.**

| FACTOR | Females (N = 18,855) | | | | Males (N = 11,628) | | | |
|---|---|---|---|---|---|---|---|---|
| | LILONGWE–URBAN | KARONGA-RURAL | TOTAL | p-value | LILONGWE–URBAN | KARONGA-RURAL | TOTAL | p-value |
| **Age group (years)** | | | | | | | | |
| 18–29 | 5,581 (51.4) | 2,952 (36.9) | 8,533 (45.3) | **<0.001** | 2,944 (50.8) | 2,224 (38.2) | 5,168 (44.4) | **<0.001** |
| 30–39 | 3,027 (27.9) | 1,980 (24.8) | 5,007 (26.6) | | 1,417 (24.4) | 1,425 (24.4) | 2,842 (24.4) | |
| 40–49 | 1,140 (10.5) | 1,232 (15.4) | 2,372 (12.6) | | 695 (12.0) | 905 (15.5) | 1,600 (13.8) | |
| 50–59 | 623 (5.7) | 835 (10.4) | 1,458 (7.7) | | 368 (6.4) | 576 (9.9) | 944 (8.1) | |
| 60–69 | 317 (2.9) | 522 (6.5) | 839 (4.5) | | 242 (4.2) | 316 (5.4) | 558 (4.8) | |
| 70+ | 171 (1.6) | 474 (5.9) | 645 (3.4) | | 132 (2.3) | 384 (6.6) | 516 (4.4) | |
| **Body Mass Index (kg/m2)** | | | | | | | | |
| Underweight (<18.5) | 418 (4.1) | 529 (7.1) | 947 (5.3) | **<0.001** | 410 (7.1) | 564 (9.7) | 974 (8.4) | **<0.001** |
| Normal (18.5–24.9) | 5298 (51.5) | 4900 (65.6) | 10198 (57.5) | | 4307 (74.3) | 4729 (81.3) | 9036 (77.8) | |
| Overweight (25.0–29.9) | 2723 (26.5) | 1465 (19.6) | 4188 (23.6) | | 855 (14.8) | 463 (8.0) | 1318 (11.4) | |
| Obese(>30.0) | 1846 (18.0) | 572 (7.7) | 2418 (13.6) | | 224 (3.9) | 59 (1.0) | 283 (2.4) | |
| **Physical activity rating** | | | | | | | | |
| Low | 178 (1.6) | 148 (1.9) | 326 (1.7) | **0.001** | 240 (4.1) | 145 (2.5) | 385 (3.3) | **<0.001** |
| Moderate | 284 (2.6) | 278 (3.5) | 562 (3.0) | | 775 (13.4) | 624 (10.7) | 1,399 (12.0) | |
| High | 10,398 (95.8) | 7,569 (94.7) | 17,967 (95.3) | | 4,783 (82.5) | 5,061 (86.8) | 9,844 (84.7) | |
| **Household income, median (IQR)** | 7 (5–9) | 4 (2–5) | 5 (3–8) | **<0.001** | 7 (5–9) | 4 (3–6) | 6 (4–8) | **<0.001** |
| **Household asset index, median (IQR)** | 7 (5–9) | 4 (2–6) | 6 (3–8) | **<0.001** | 7 (5–9) | 4 (3–7) | 6 (3–8) | **<0.001** |
| **Education** | | | | | | | | |
| No formal | 505 (4.7) | 496 (6.2) | 1,001 (5.3) | **<0.001** | 78 (1.4) | 99 (1.7) | 177 (1.5) | **<0.001** |
| Primary: Standard 1–5 | 1,059 (9.8) | 1,366 (17.1) | 2,425 (12.9) | | 272 (4.7) | 634 (10.9) | 906 (7.8) | |
| Primary: Standard 6–8 | 2,486 (22.9) | 4,065 (50.8) | 6,551 (34.7) | | 814 (14.0) | 2,407 (41.3) | 3,221 (27.7) | |
| Secondary | 5,357 (49.3) | 1,990 (24.9) | 7,347 (39.0) | | 3,333 (57.5) | 2,515 (43.1) | 5,848 (50.3) | |
| Tertiary | 1,453 (13.4) | 78 (1.0) | 1,531 (8.1) | | 1,301 (22.4) | 175 (3.0) | 1,476 (12.7) | |
| **Work Status** | | | | | | | | |
| Not working | 2,525 (23.3) | 730 (9.1) | 3,255 (17.3) | **<0.001** | 2,085 (36.0) | 1,012 (17.4) | 3,097 (26.6) | **<0.001** |
| Housework | 4,619 (42.5) | 763 (9.5) | 5,382 (28.5) | | 364 (6.3) | 70 (1.2) | 434 (3.7) | |
| Farming/fishing | 57 (0.5) | 5,266 (65.9) | 5,323 (28.2) | | 30 (0.5) | 3,369 (57.8) | 3,399 (29.2) | |
| Self-employed | 1,957 (18.0) | 1,040 (13.0) | 2,997 (15.9) | | 1,163 (20.1) | 852 (14.6) | 2,015 (17.3) | |
| Employed | 1,702 (15.7) | 196 (2.5) | 1,898 (10.1) | | 2,156 (37.2) | 527 (9.0) | 2,683 (23.1) | |
| **Diagnosed Diabetes Mellitus** | | | | | | | | |

*(Continued)*

**Table 1.** (Continued)

| FACTOR | Females (N = 18,855) | | | | Males (N = 11,628) | | | |
|---|---|---|---|---|---|---|---|---|
| | LILONGWE–URBAN | KARONGA-RURAL | TOTAL | p-value | LILONGWE–URBAN | KARONGA-RURAL | TOTAL | p-value |
| No | 10704 (98.6) | 7926 (99.1) | 18630 (98.8) | <**0.001** | 5706 (98.4) | 5789 (99.3) | 11495 (98.9) | <**0.001** |
| Yes | 156 (1.4) | 69 (0.9) | 225 (1.2) | | 92 (1.6) | 41 (0.7) | 133 (1.1) | |
| **Diagnosed Heart Disease** | | | | | | | | |
| No | 10578 (97.4) | 7825 (97.9) | 18403 (97.6) | **0.037** | 5739 (99.0) | 5784 (99.2) | 11523 (99.1) | 0.193 |
| Yes | 282 (2.6) | 170 (2.1) | 452 (2.4) | | 59 (1.0) | 46 (0.8) | 105 (0.9) | |
| **Diagnosed High Blood Pressure** | | | | | | | | |
| No | 9448 (87.0) | 7339 (91.8) | 16787 (89.0) | <**0.001** | 5337 (92.1) | 5606 (96.2) | 10943 (94.1) | <**0.001** |
| Yes | 1412 (13.0) | 656 (8.2) | 2068 (11.0) | | 461 (8.0) | 224 (3.8) | 685 (5.9) | |
| **Diagnosed Stroke** | | | | | | | | |
| No | 10784 (99.3) | 7941 (99.3) | 18725 (99.3) | 0.841 | 5750 (99.2) | 5782 (99.2) | 11532 (99.2) | 0.978 |
| Yes | 76 (0.7) | 54 (0.7) | 130 (0.7) | | 48 (0.8) | 48 (0.8) | 96 (0.8) | |
| **Piped water at home** | | | | | | | | |
| No | 6,906 (63.6) | 7,404 (92.6) | 14,310 (75.9) | <**0.001** | 3,404 (58.7) | 5,345 (91.7) | 8,749 (75.2) | <**0.001** |
| Yes | 3,954 (36.4) | 591 (7.4) | 4,545 (24.1) | | 2,394 (41.3) | 485 (8.3) | 2,879 (24.8) | |
| **Smoking Status** | | | | | | | | |
| Never smoked | 10,790 (99.4) | 7,971 (99.7) | 18,761 (99.5) | **0.001** | 4,850 (83.7) | 4,777 (81.9) | 9,627 (82.8) | **0.001** |
| Former smoker (stopped more than 6 months ago) | 43 (0.4) | 10 (0.1) | 53 (0.3) | | 442 (7.6) | 262 (4.5) | 704 (6.1) | |
| Current (in last 6 months) | 27 (0.3) | 14 (0.2) | 41 (0.2) | | 506 (8.7) | 791 (13.6) | 1,297 (11.2) | |
| **Smoker in household** | | | | | | | | |
| No | 9,722 (89.5) | 6,755 (84.5) | 16,477 (87.4) | <**0.001** | 5,305 (91.5) | 5,249 (90.0) | 10,554 (90.8) | **0.006** |
| Yes | 1,138 (10.5) | 1,240 (15.5) | 2,378 (12.6) | | 493 (8.5) | 581 (10.0) | 1,074 (9.2) | |
| **Firewood smoke exposure** | | | | | | | | |
| no/former/little exposure | 1003 (9.4) | 266 (3.4) | 1269 (6.8) | <**0.001** | 1240 (21.8) | 2392 (43.6) | 3632 (32.5) | <**0.001** |
| exposed, ventilated cooking area | 8474 (79.3) | 7541 (96.0) | 16015 (86.3) | | 3874 (68.1) | 3073 (56.0) | 6947 (62.1) | |
| exposed non-ventilated cooking area | 1216 (11.4) | 49 (0.6) | 1265 (6.8) | | 576 (10.1) | 24 (0.4) | 600 (5.4) | |

• P-values were calculated by using Pearson χ2 test with 1 degree of freedom or Mann Whitney U test for continuous variables.

• Values in boldface indicate P<0.05

• Missing data for

 ○ **Age group:** 1 urban female participant

 ○ **Body Mass Index:** 544 rural and 577 urban participants

 ○ **Firewood smoke exposure:** 490 rural and 275 urban participants

**Table 2. Asthma outcomes and symptoms stratified by sex and urban-rural area.**

| Asthma Outcomes | Females (N = 18,855) | | | | Males (N = 11,628) | | | |
|---|---|---|---|---|---|---|---|---|
| | LILONGWE–URBAN | KARONGA-RURAL | TOTAL | p-value | LILONGWE–URBAN | KARONGA–RURAL | TOTAL | p-value |
| **Diagnosed asthma (Yes)** | **611 (5.6)** | **362 (4.5)** | **973 (5.2)** | **0.001** | **253 (4.4)** | **258 (4.4)** | **511 (4.4)** | **0.871** |
| Age of asthma diagnosis (years), median (IQR) subset | 14.0 (5.0–23.0) | 17.0 (6.5–28.0) | 15.0 (5.0–25.0) | **<0.001** | 11.0 (4.0–19.8) | 12.0 (5.0–25.0) | 12.0 (5.0–22.0) | 0.073 |
| Diagnosed asthma with medication (Yes) | 256 (2.4) | 186 (2.3) | 442 (2.3) | 0.890 | 82 (1.4) | 101 (1.7) | 183 (1.6) | 0.168 |
| Wheeze in the past 12 months (Yes) | 371 (3.4) | 257 (3.2) | 628 (3.3) | 0.446 | 168 (2.9) | 193 (3.3) | 361 (3.1) | 0.199 |
| Wheeze in the past 12 months plus shortness of breath (Yes) | 501 (4.6) | 262 (3.3) | 763 (4.1) | **<0.001** | 237 (4.1) | 177 (3.0) | 414 (3.6) | **0.002** |
| Asthma plus Wheeze in the past 12 months (Yes) | 135 (1.2) | 101 (1.3) | 236 (1.3) | 0.902 | 46 (0.8) | 66 (1.1) | 112 (1.0) | 0.062 |
| Asthma diagnosed < 18 years (Yes) | 340 (59.9) | 147 (50.2) | 487 (56.6) | **0.007** | 166 (71.6) | 135 (61.6) | 301 (66.7) | **0.026** |

• P-values were calculated by using Pearson χ2 test with 1 degree of freedom or Mann Whitney U test for continuous variables.

• Values in boldface indicate P<0.05

diagnosed blood pressure, piped water at home, smoking status, and firewood smoke exposure.

**Demographic, anthropometric, and socioeconomic factors.** As shown in Table 3, after adjusting for other factors, and considering a Bonferroni corrected significance level of p = 0.004, in urban females only, being obese was associated with increased odds of diagnosed asthma compared to being normal weight (OR = 1.59, 95%CI 1.26–2.01, p<0.001). In rural females, a secondary school education (OR = 2.10, 95%CI 1.35–3.27, p = 0.001) was positively associated with diagnosed asthma compared to completing primary Standard 1–5. By contrast, urban females completing primary Standard 6–8 had reduced odds of reported diagnosed asthma (OR = 0.58, 95%CI 0.41–0.81, p = 0.001) compared to those completing primary Standard 1–5. No significant associated were observed for education and asthma when it came to the male groups.

**Other NCD comorbidities.** We observed that previously diagnosed heart disease (yes versus no) was positively associated diagnosed asthma in rural females (OR = 2.30, 95%CI 1.32–4.02, p = 0.003). A similar observation was made in urban females, but this was not significant if we considered multiple testing. No association was seen in either male group with regards to any NCD comorbidity.

**Environmental factors–water source, smoke exposure.** In urban females, being a former smoker compared to having never smoked, was associated with diagnosed asthma (OR = 4.17, 95%CI [1.94–8.94], p<0.001). However, the number of females who were smokers/former smokers in this population was relatively small. In rural males only, being a current smoker (in the last 6 months) compared to having never smoked had an inverse association with asthma diagnosis (OR = 0.46, 95% CI [0.27–0.79], p = 0.004).

## Factors associated with wheeze in the past 12 months

S2 Table shows the results of univariable logistic regression analysis of potential factors associated with reported wheeze in the past 12 months stratified by sex and urban-rural area. All factors found to be associated with wheeze (p<0.100) were explored further in multivariable analysis. The factors included in the final multivariable model were age-group, BMI, physical activity rating, household income, household asset index, education, work status, diagnosed

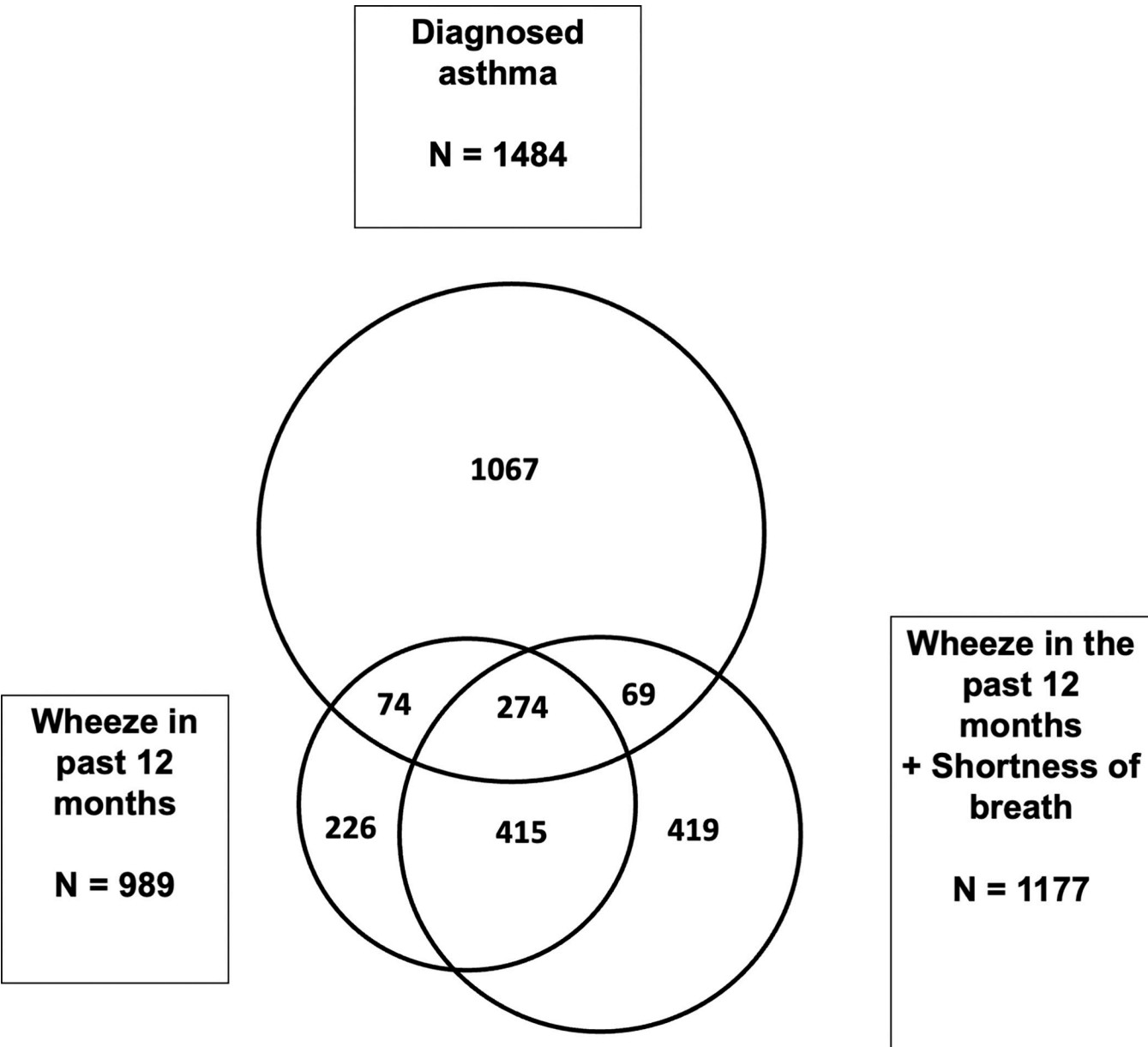

**Fig 1. Overlap between 'Diagnosed asthma" and related symptoms.** Figure is a proportional Venn diagram showing the overlap between the outcomes 'Diagnosed asthma', 'Wheeze in the past 12 months' and 'Wheeze in the past 12 months plus shortness of breath'. The area representing each variable is proportional to the number of people reporting this outcome.

heart disease, diagnosed blood pressure, diagnosed stroke, piped water at home, smoking status, and firewood smoke exposure.

**Demographic, anthropometric, and socioeconomic factors.** No association was observed between being obese (compared to normal weight) and 'wheeze in the past 12 months' in urban females (Table 4). Increasing household income as a linear variable was significantly associated with wheeze in the past 12 months among rural males (OR = 1.13, 95%CI 1.06–1.22, p<0.001). When it came to household possessions, increasing 'household asset index' was positively associated with 'wheeze in the past 12 months' in urban females only

**Table 3. Multivariable analysis—Factors associated with diagnosed asthma stratified by sex & urban-rural area.**

| Factors | Females | | | | Males | | | |
|---|---|---|---|---|---|---|---|---|
| | LILONGWE—URBAN | | KARONGA—RURAL | | LILONGWE—URBAN | | KARONGA—RURAL | |
| | Adjusted Odds ratio (95%CI) | p-value | Adjusted Odds ratio (95%CI) | p-value | Adjusted Odds ratio (95%CI) | p-value | Adjusted Odds ratio (95%CI) | p-value |
| **Age group** | | | | | | | | |
| 18–29 | 1.00 | - | 1.00 | - | 1.00 | - | 1.00 | - |
| 30–39 | 1.14 (0.92–1.41) | 0.225 | 1.43 (1.06–1.92) | 0.020 | 0.99 (0.69–1.43) | 0.958 | 1.14 (0.81–1.61) | 0.439 |
| 40–49 | 0.94 (0.69–1.28) | 0.685 | 1.34 (0.94–1.91) | 0.102 | 1.03 (0.66–1.62) | 0.883 | 0.63 (0.39–1.00) | 0.051 |
| 50–59 | 1.42 (0.98–2.04) | 0.063 | 1.05 (0.67–1.65) | 0.843 | 0.75 (0.40–1.39) | 0.364 | 1.02 (0.62–1.67) | 0.945 |
| 60–69 | 1.17 (0.69–1.99) | 0.556 | 1.92 (1.17–3.15) | 0.010 | 0.59 (0.27–1.31) | 0.199 | 0.85 (0.43–1.68) | 0.637 |
| 70+ | 1.51 (0.74–3.08) | 0.259 | 0.84 (0.39–1.83) | 0.663 | 0.56 (0.19–1.69) | 0.307 | 1.13 (0.58–2.19) | 0.727 |
| **BMI (WHO categories)** | | | | | | | | |
| Underweight (<18.5) | 0.93 (0.57–1.52) | 0.775 | 1.26 (0.83–1.92) | 0.276 | 1.08 (0.66–1.76) | 0.750 | 1.52 (1.02–2.27) | 0.040 |
| Normal(18.5–24.9) | 1.00 | - | 1.00 | - | 1.00 | - | 1.00 | - |
| Overweight(25.0–29.9) | 1.30 (1.05–1.60) | 0.014 | 0.91 (0.68–1.24) | 0.561 | 0.91 (0.62–1.34) | 0.647 | 0.93 (0.57–1.53) | 0.786 |
| Obese(>30) | **1.59 (1.26–2.01)** | **<0.001** | 1.20 (0.81–1.80) | 0.367 | 0.71 (0.35–1.47) | 0.359 | 1.05 (0.32–3.49) | 0.933 |
| **Physical activity rating** | | | | | | | | |
| Low | 1.00 | - | 1.00 | - | 1.00 | - | 1.00 | - |
| Moderate | 0.71 (0.33–1.51) | 0.367 | 1.01 (0.29–3.51) | 0.987 | 1.10 (0.56–2.15) | 0.783 | 0.31 (0.14–0.72) | 0.007 |
| High | 0.73 (0.40–1.36) | 0.321 | 1.56 (0.52–4.73) | 0.430 | 0.74 (0.39–1.41) | 0.368 | 0.56 (0.27–1.12) | 0.102 |
| **Household income** | 1.00 (0.97–1.04) | 0.800 | 1.00 (0.95–1.06) | 0.966 | 1.03 (0.97–1.09) | 0.322 | 1.08 (1.01–1.15) | 0.017 |
| **Household asset index** | 1.04 (0.99–1.08) | 0.094 | 0.99 (0.94–1.04) | 0.667 | 1.05 (0.98–1.12) | 0.149 | 1.02 (0.96–1.08) | 0.528 |
| **Education** | | | | | | | | |
| No formal | 0.52 (0.30–0.91) | 0.022 | 1.19 (0.64–2.22) | 0.584 | 4.07 (0.98–16.82) | 0.053 | 0.50 (0.07–3.88) | 0.510 |
| Primary: Standard 1–5 | 1.00 | - | 1.00 | - | 1.00 | - | 1.00 | - |
| Primary: Standard 6–8 | **0.58 (0.41–0.81)** | **0.001** | 1.68 (1.14–2.48) | 0.009 | 2.73 (0.96–7.82) | 0.061 | 1.48 (0.84–2.60) | 0.170 |
| Secondary | 0.93 (0.69–1.26) | 0.648 | **2.10 (1.35–3.27)** | **0.001** | 2.43 (0.87–6.75) | 0.089 | 1.74 (0.98–3.07) | 0.058 |
| Tertiary | 1.44 (1.00–2.08) | 0.048 | 2.69 (0.99–7.32) | 0.053 | 3.39 (1.19–9.67) | 0.023 | 1.86 (0.80–4.32) | 0.150 |
| **Work Status** | | | | | | | | |
| Not working | 1.00 | - | 1.00 | - | 1.00 | - | 1.00 | - |
| Housework | 1.09 (0.85–1.40) | 0.486 | 0.76 (0.45–1.28) | 0.294 | 1.08 (0.62–1.89) | 0.790 | 2.41 (1.03–5.66) | 0.043 |
| Farming/fishing | 0.75 (0.18–3.20) | 0.703 | 0.77 (0.51–1.17) | 0.225 | 0.86 (0.11–6.57) | 0.885 | 0.93 (0.64–1.36) | 0.703 |
| Self-employed | 1.32 (1.00–1.76) | 0.054 | 0.85 (0.53–1.38) | 0.518 | 1.13 (0.75–1.70) | 0.548 | 0.77 (0.47–1.25) | 0.285 |
| Employed | 1.19 (0.90–1.58) | 0.228 | 1.23 (0.60–2.52) | 0.572 | 0.76 (0.52–1.10) | 0.145 | 1.16 (0.69–1.95) | 0.573 |
| **Diagnosed Heart Disease** | | | | | | | | |
| No | 1.00 | - | 1.00 | - | 1.00 | - | 1.00 | - |
| Yes | 1.76 (1.16–2.68) | 0.008 | **2.30 (1.32–4.02)** | **0.003** | 1.17 (0.36–3.84) | 0.792 | 0.96 (0.23–4.04) | 0.952 |
| **Diagnosed High Blood Pressure** | | | | | | | | |
| No | 1.00 | - | 1.00 | - | 1.00 | - | 1.00 | - |
| Yes | 0.93 (0.72–1.21) | 0.611 | 1.22 (0.80–1.85) | 0.357 | 1.16 (0.70–1.92) | 0.567 | 0.98 (0.49–1.95) | 0.951 |
| **Piped water at home** | | | | | | | | |
| No | 1.00 | - | 1.00 | - | 1.00 | - | 1.00 | - |
| Yes | 1.05 (0.85–1.29) | 0.658 | 1.23 (0.84–1.82) | 0.286 | 0.92 (0.67–1.26) | 0.601 | 1.18 (0.78–1.81) | 0.431 |
| **Smoking Status** | | | | | | | | |
| Never smoked | 1.00 | | 1.00 | | 1.00 | | 1.00 | |
| Former smoker (stopped more than 6 months ago) | 4.17 (1.94–8.94) | <0.001 | *no output* | - | 1.06 (0.66–1.70) | 0.822 | 1.09 (0.59–2.00) | 0.787 |
| Current (in last 6 months) | 0.81 (0.11–6.09) | 0.839 | *no output* | - | 0.85 (0.52–1.41) | 0.534 | **0.46 (0.27–0.79)** | **0.004** |
| **Firewood smoke exposure** | | | | | | | | |

*(Continued)*

**Table 3.** (Continued)

| Factors | Females | | | | Males | | | |
|---|---|---|---|---|---|---|---|---|
| | LILONGWE—URBAN | | KARONGA—RURAL | | LILONGWE—URBAN | | KARONGA—RURAL | |
| | Adjusted Odds ratio (95%CI) | p-value | Adjusted Odds ratio (95%CI) | p-value | Adjusted Odds ratio (95%CI) | p-value | Adjusted Odds ratio (95%CI) | p-value |
| No/former/little exposure | 1.00 | - | 1.00 | - | 1.00 | - | 1.00 | - |
| Exposed, ventilated cooking area | 1.41 (1.04–1.90) | 0.027 | 0.65 (0.35–1.23) | 0.189 | 0.81 (0.60–1.11) | 0.189 | 1.00 (0.76–1.32) | 0.993 |
| Exposed, non-ventilated cooking area | 1.07 (0.72–1.60) | 0.733 | 0.72 (0.15–3.43) | 0.684 | 0.64 (0.37–1.11) | 0.113 | 0.96 (0.13–7.27) | 0.966 |

• **Final model adjusted for**: age-group, BMI, physical activity rating, household income, household asset index, education, work status, diagnosed heart disease, diagnosed high blood pressure, piped water at home, smoking status, and firewood smoke exposure

• Adjusted ORs, 95% CIs and p-values in boldface indicate model results where $p \leq 0.004$ (significance level threshold after Bonferroni correction)

• *no output*: models with no output for this level due to small numbers

(OR = 1.09, 95%CI 1.03–1.15, p = 0.003). Unlike with self-diagnosed asthma, educational status was not associated with wheeze in the past 12 months.

**Other NCD comorbidities.** In all groups, reported diagnosed heart disease was significantly associated with wheeze in the past 12 months. This was observed for rural females (OR = 2.37, 95%CI 1.32–4.26, p = 0.004), urban females (OR = 4.84, 95%CI 3.34–7.00, p<0.001), urban males (OR = 5.59, 95%CI 2.49–12.57, p<0.001) and rural males (OR = 4.76, 95%CI 2.15–10.56, p<0.001). We also observed that a prior diagnosis of high blood pressure was associated with reported wheeze in the past 12 months in rural females only (OR = 2.24, 95%CI 1.50–3.33, p<0.001).

**Environmental factors–water source, smoke exposure.** There was no strong evidence for an association between smoking status and wheeze outcomes. Exposure to firewood in ventilated cooking area was associated with reported wheeze in the past 12 months in rural males only but this did not reach significance after correcting for multiple testing (OR = 1.55, 95%CI 1.12–2.14, p = 0.008).

## Discussion

In this large community-based survey of Malawian adults, we observed only a slightly higher prevalence of diagnosed asthma in urban compared to rural dwelling adults which is in contrast to findings from elsewhere in Sub-Saharan Africa where asthma has been strongly linked to urbanisation and the adoption of a so-called western lifestyle [17–19, 21]. It is important to note that differences in underlying socioeconomic structures between African countries make direct comparisons challenging. For example, in Malawi, universally constrained access to healthcare and diagnostics may contribute to the lack of distinct urban-rural differences. In addition, the socioeconomic distribution of our study population may not be comparable to other studies that may have included urban dwellers of higher socioeconomic status with greater access to health-care [43, 44].

In our analysis we stratified our findings by both sex and urban-rural study area. The rationale for this was that published findings from our population cohort observed differences in the prevalence of disease outcomes such as hypertension according to urban-rural study area as well as sex [34]. Additionally, our preliminary analysis indicated that some sociodemographic indicators varied by both urban-rural area and sex.

We observed poor overlap between diagnosed asthma and asthma symptoms (reported wheeze in the past 12 months and wheeze in the past 12 months plus shortness of breath). This

**Table 4. Multivariable analysis—Factors associated with 'Wheeze in the past 12 months' stratified by sex & urban-rural area.**

| Factors | Female | | | | Male | | | |
|---|---|---|---|---|---|---|---|---|
| | LILONGWE–URBAN | | KARONGA—RURAL | | LILONGWE—URBAN | | KARONGA—RURAL | |
| | Adjusted Odds ratio (95%CI) | p-value | Adjusted Odds ratio (95%CI) | p-value | Adjusted Odds ratio (95%CI) | p-value | Adjusted Odds ratio (95%CI) | p-value |
| **Age group** | | | | | | | | |
| 18–29 | 1.00 | - | 1.00 | - | 1.00 | - | 1.00 | - |
| 30–39 | 0.98 (0.74–1.29) | 0.874 | 1.32 (0.90–1.93) | 0.157 | 1.14 (0.72–1.79) | 0.575 | 0.98 (0.65–1.48) | 0.933 |
| 40–49 | 1.00 (0.68–1.47) | 0.992 | 1.56 (1.03–2.36) | 0.037 | 1.17 (0.67–2.05) | 0.581 | 1.22 (0.77–1.92) | 0.398 |
| 50–59 | 1.35 (0.87–2.08) | 0.180 | 1.41 (0.87–2.30) | 0.165 | 1.56 (0.82–2.96) | 0.175 | 0.83 (0.44–1.56) | 0.567 |
| 60–69 | 0.61 (0.30–1.22) | 0.160 | 1.71 (0.98–3.00) | 0.060 | 1.56 (0.71–3.43) | 0.264 | 1.55 (0.79–3.01) | 0.200 |
| 70+ | 1.01 (0.43–2.36) | 0.982 | 0.63 (0.28–1.42) | 0.262 | 1.57 (0.55–4.53) | 0.401 | 1.34 (0.64–2.83) | 0.439 |
| **BMI (WHO categories)** | | | | | | | | |
| Underweight (<18.5) | 1.06 (0.60–1.90) | 0.832 | 1.18 (0.71–1.96) | 0.528 | 1.25 (0.72–2.18) | 0.422 | 1.25 (0.77–2.03) | 0.358 |
| Normal(18.5–24.9) | 1.00 | - | 1.00 | - | 1.00 | - | 1.00 | - |
| Overweight(25.0–29.9) | 1.25 (0.96–1.62) | 0.094 | 1.00 (0.71–1.40) | 0.991 | 0.65 (0.39–1.09) | 0.104 | 1.45 (0.89–2.34) | 0.134 |
| Obese(>30) | 1.15 (0.85–1.57) | 0.358 | 1.19 (0.76–1.85) | 0.445 | 0.58 (0.22–1.51) | 0.266 | 2.11 (0.73–6.15) | 0.169 |
| **Physical activity rating** | | | | | | | | |
| Low | 1.00 | - | 1.00 | - | 1.00 | - | 1.00 | - |
| Moderate | 1.77 (0.76–4.12) | 0.189 | 0.80 (0.27–2.34) | 0.680 | 2.61 (0.97–7.01) | 0.058 | 0.42 (0.17–1.07) | 0.068 |
| High | 0.81 (0.37–1.75) | 0.589 | 0.82 (0.31–2.18) | 0.695 | 1.76 (0.66–4.65) | 0.257 | 0.64 (0.29–1.44) | 0.284 |
| **Household income** | 1.02 (0.98–1.07) | 0.303 | 1.03 (0.96–1.09) | 0.434 | 1.08 (1.00–1.16) | 0.045 | **1.13 (1.06–1.22)** | **<0.001** |
| **Household asset index** | **1.09 (1.03–1.15)** | **0.003** | 1.02 (0.97–1.08) | 0.392 | 1.01 (0.94–1.09) | 0.792 | 0.96 (0.89–1.02) | 0.184 |
| **Education** | | | | | | | | |
| No formal | 0.69 (0.37–1.30) | 0.256 | 1.14 (0.62–2.06) | 0.677 | *no output* | - | 1.11 (0.25–5.03) | 0.892 |
| Primary: Standard 1–5 | 1.00 | - | 1.00 | - | 1.00 | - | 1.00 | - |
| Primary: Standard 6–8 | 0.64 (0.43–0.96) | 0.030 | 0.98 (0.66–1.44) | 0.905 | 1.26 (0.57–2.76) | 0.570 | 1.25 (0.68–2.29) | 0.472 |
| Secondary | 0.73 (0.50–1.06) | 0.101 | 1.07 (0.66–1.72) | 0.782 | 0.85 (0.40–1.82) | 0.680 | 1.59 (0.86–2.93) | 0.138 |
| Tertiary | 0.85 (0.54–1.35) | 0.489 | 1.67 (0.57–4.93) | 0.349 | 0.98 (0.43–2.23) | 0.957 | 0.88 (0.30–2.58) | 0.821 |
| **Work Status** | | | | | | | | |
| Not working | | | | | | | | |
| Housework | 0.79 (0.58–1.08) | 0.143 | 0.47 (0.24–0.91) | 0.025 | 0.16 (0.04–0.65) | 0.011 | 0.85 (0.20–3.69) | 0.829 |
| Farming/fishing | 3.12 (1.23–7.90) | 0.016 | 0.66 (0.41–1.06) | 0.086 | 0.96 (0.12–7.40) | 0.965 | 0.96 (0.61–1.49) | 0.841 |
| Self-employed | 0.98 (0.69–1.39) | 0.917 | 0.81 (0.46–1.41) | 0.455 | 0.91 (0.56–1.47) | 0.690 | 0.85 (0.49–1.47) | 0.559 |
| Employed | 0.84 (0.59–1.20) | 0.338 | 1.26 (0.56–2.82) | 0.579 | 0.65 (0.42–1.01) | 0.055 | 1.10 (0.60–2.01) | 0.762 |
| **Diagnosed heart disease** | | | | | | | | |
| No | 1.00 | - | 1.00 | - | 1.00 | - | 1.00 | - |
| Yes | **4.84 (3.34–7.00)** | **<0.001** | **2.37 (1.32–4.26)** | **0.004** | **4.76 (2.15–10.56)** | **<0.001** | **5.59 (2.49–12.57)** | **<0.001** |
| **Diagnosed High Blood Pressure** | | | | | | | | |
| No | 1.00 | - | 1.00 | - | 1.00 | - | 1.00 | - |
| Yes | 1.30 (0.96–1.76) | 0.096 | **2.24 (1.50–3.33)** | **<0.001** | 0.71 (0.37–1.36) | 0.300 | 1.45 (0.73–2.90) | 0.287 |
| **Diagnosed Stroke** | | | | | | | | |
| No | 1.00 | - | 1.00 | - | 1.00 | - | 1.00 | - |
| Yes | 0.82 (0.28–2.39) | 0.710 | 1.36 (0.40–4.66) | 0.621 | 1.51 (0.34–6.68) | 0.590 | 1.06 (0.24–4.72) | 0.938 |
| **Piped water at home** | | | | | | | | |
| No | 1.00 | - | 1.00 | - | 1.00 | - | 1.00 | - |
| Yes | 0.99 (0.77–1.29) | 0.955 | 0.76 (0.45–1.27) | 0.291 | 0.92 (0.63–1.35) | 0.661 | 0.97 (0.58–1.64) | 0.915 |
| **Smoking status** | | | | | | | | |
| Never smoked | 1.00 | - | 1.00 | - | | | | |

*(Continued)*

**Table 4.** (Continued)

| Factors | Female | | | | Male | | | |
|---|---|---|---|---|---|---|---|---|
| | LILONGWE–URBAN | | KARONGA—RURAL | | LILONGWE—URBAN | | KARONGA—RURAL | |
| | Adjusted Odds ratio (95%CI) | p-value | Adjusted Odds ratio (95%CI) | p-value | Adjusted Odds ratio (95%CI) | p-value | Adjusted Odds ratio (95%CI) | p-value |
| Former smoker (stopped more than 6 months ago) | 1.06 (0.25–4.59) | 0.933 | *no output* | - | 1.16 (0.67–2.01) | 0.605 | 1.27 (0.67–2.43) | 0.463 |
| Current (in last 6 months) | 1.43 (0.19–10.72) | 0.731 | 8.00 (1.57–40.90) | 0.012 | 1.23 (0.73–2.08) | 0.428 | 1.21 (0.78–1.86) | 0.401 |
| **Firewood smoke exposure** | | | | | | | | |
| No/former/little exposure | 1.00 | - | 1.00 | - | 1.00 | - | 1.00 | - |
| Exposed, ventilated cooking area | 1.02 (0.73–1.44) | 0.900 | 0.90 (0.44–1.84) | 0.772 | 0.67 (0.46–0.97) | 0.033 | 1.55 (1.12–2.14) | 0.008 |
| Exposed, non-ventilated cooking area | 0.88 (0.55–1.39) | 0.575 | *no output* | - | 1.06 (0.62–1.82) | 0.820 | No output | - |

• **Final model adjusted for**: age-group, BMI, physical activity rating, household income, household asset index, education, work status, diagnosed heart disease, diagnosed high blood pressure, diagnosed stroke, piped water at home, smoking status, and firewood smoke exposure.

• Adjusted ORs, 95% CIs and p-values in boldface indicate model results where p≤0.004 (significance level threshold after Bonferroni correction)

• *no output*: models with no output for this level due to small numbers

could indicate that most participants who had previously been diagnosed with asthma were asymptomatic within the year prior to the questionnaire interview. It is also possible that reported wheeze may not be related to diagnosed asthma in our population. Additionally, other studies from Malawi have found a notable burden of abnormal lung function due to other chronic respiratory conditions such as chronic obstructive pulmonary disorder (COPD) [29, 30, 32].

The older age of asthma diagnosis in rural compared to urban dwellers that we noted could reflect a true higher prevalence of adult-onset asthma in the rural area. Nonetheless, it may also reflect some differences in health-seeking behaviours and improved access to healthcare in our rural area over time. We also observed an association between self-reported diagnosed heart disease and asthma which is consistent with other investigations [45–47]. Similar to our own investigation, other studies report that the association is stronger among adult females compared to males [47]. Although the underlying mechanisms linking asthma and heart disease, particularly in females, remain unclear, an inflammatory cytokine milieu and oestrogen-modulated inflammation may play roles [48]. We also found an association between wheeze in the past 12 months and diagnosed heart disease in all groups. Due to lack of access to diagnostics within health centres in Malawi, an asthma and heart disease diagnoses are based mainly on clinical observations by health-care professionals and symptoms that may be similar. Hence some misclassification of these conditions is inevitable [49].

We did not observe a link between diagnosed high blood pressure and reported asthma in any group although a link between these two conditions has been established [50]. Interestingly, we did observe an association between diagnosed high blood pressure and reported wheeze in the past 12 months in rural females. However, this observation may not be linked to asthma given the lack of overlap between asthma and wheeze in our study.

Our observed strong association between obesity (BMI>30kg/m$^2$), and diagnosed asthma in urban females has been seen in other parts of the world [51, 52]. Although obesity is thought to affect the metabolic and immunological pathways that lead to asthma, specific mechanisms involved remain unclear [53]. One possible pathway is through the action of adipokines (cytokines produced by adipose tissue in obese individuals), which may affect the inflammatory process in the lungs and therefore asthma pathogenesis [53]. Nonetheless, the association

between excess adiposity and asthma was not observed in urban males. This might reflect a female predisposition to adult-onset asthma, as observed in other parts of the world [54, 55]. However, we cannot rule out a potential misdiagnosis of asthma in obese individuals since both conditions are characterised by breathlessness [56].

In our study population, the relationship between socioeconomic status (SES) and diagnosed asthma was not consistent. This could reflect the limitations of our SES assessment tools or an indication that a clear relationship did not exist. We did however observe that higher education appeared to be associated with asthma in urban females. It is possible that educational level was a better proxy for some of the socioeconomic factors associated with increased risk of asthma and this was apparent in urban females because of the high numbers of this group in our study. It could also reflect more health-seeking behaviour in urban females compared to their rural counterparts or better access to health-care facilities in an urban setting compared to a rural environment.

When we examined physical activity, we observed a negative association between moderate physical activity (compared to low activity) on diagnosed asthma in rural males but this did not reach significance after taking into account multiple testing.

Regarding smoking in our study, few females, especially in the rural area, were current smokers / former smokers which reflects the cultural norms in Malawi and has also been observed in other studies [29, 57]. However, we did observe that being a former smoker was associated with diagnosed asthma in urban females which may indicate smoking cessation at the time of diagnosis and therefore be an example of reverse causation [58]. At the same time, this finding could also reflect social desirability effects since individuals with asthma may underreport some behaviours such as smoking due to perceived or actual social disapproval [59]. However, our observations regarding smoking in female participants should be interpreted with caution given the small numbers. Our study findings also pointed to a 'protective effect' of current smoking on diagnosed asthma in rural males which again may indicate that diagnosed asthmatics were less likely to smoke or were potentially underreporting their smoking habits.

We did not observe any links between smoking and wheeze in our study. Other studies conducted in Malawi have looked at the effect of smoking on respiratory symptoms (cough, phlegm, wheeze or dyspnoea) and have observed a positive associations between being a current or ex-smoker and cough (without a cold) in urban adults in Blantyre, Malawi [29] as well as between 'smoking ever' and cough (but not wheeze) in rural adults in Chikwawa District [60].

Indoor solid biomass (crop waste, dung, firewood, charcoal etc) fuel smoke exposure has been associated with increased risk of respiratory conditions in LMICs [61, 62]. Although indoor biomass fuel exposure has been linked to COPD risk [63], the relationship with asthma is unclear [64]. In our investigation we found no significant association between exposure to firewood in a non-ventilated cooking area and diagnosed asthma. We indications of a positive relationship between this exposure and wheeze in the past 12 months among rural males. Given the poor overlap between asthma and wheeze in our investigation, this observation may not be related to asthma.

The strengths of our study include the use of a standard well-validated questionnaire and having a large sample size that enabled multivariable analysis as well as urban versus rural comparisons. Our study had some limitations including our reliance on self-reported diagnosis of asthma, wheeze and disease onset which may have resulted in some misclassification and potential information bias. Additionally, clinical assessment of asthma using spirometry was not available and we were not able to differentiate between asthma, COPD or even heart failure [49]. Therefore, there will be some attenuation of results due to misclassification. Moreover,

there are likely to be some unmeasured confounders which may explain our findings. Lastly, the cross-sectional nature of the study means that we cannot mitigate the effects of reverse-causality. Hence, observed associations should be interpreted with caution.

## Conclusions

Despite limitations, our findings highlight relationships between excess body weight as well as comorbidities such as reported heart disease and diagnosed asthma in Malawi among females. Future investigations using longitudinally collected data and clinical measures for asthma diagnosis are needed to better understand these observations and to inform healthcare delivery services for asthma in urban and rural Malawi.

## Supporting information

**S1 Fig. Overlap between 'Diagnosed asthma with medication use' and related symptoms.** Figure is a proportional Venn diagram showing the overlap between the outcomes 'Diagnosed asthma with medication use', 'Wheeze in the past 12 months' and 'Wheeze in the past 12 months plus shortness of breath'. The area representing each variable is proportional to the number of people reporting this outcome.
(TIFF)

**S1 Table. Univariable analysis—Factors associated with diagnosed asthma stratified by sex & urban-rural area.**
(DOCX)

**S2 Table. Univariable analysis—Factors associated with wheeze in the past 12 months stratified by sex & urban-rural area.**
(DOCX)

## Acknowledgments

We are very grateful to study participants and community leaders for taking part in our investigation. We are also thankful to Malawi Epidemiology and Intervention Research Unit (previously known as Karonga Prevention Study) research clinicians, interviewers, research assistants and all other staff who were part of gathering data for the study.

## Author Contributions

**Conceptualization:** Alison J. Price, Alemayehu Amberbir, Amelia C. Crampin.

**Data curation:** Estelle McLean, Amelia C. Crampin.

**Formal analysis:** Abena S. Amoah.

**Funding acquisition:** Amelia C. Crampin.

**Investigation:** Alison J. Price, Alemayehu Amberbir, Amelia C. Crampin.

**Project administration:** Amelia C. Crampin.

**Supervision:** Estelle McLean, Alison J. Price, Alemayehu Amberbir, Amelia C. Crampin.

**Writing – original draft:** Abena S. Amoah.

**Writing – review & editing:** Abena S. Amoah, Estelle McLean, Alison J. Price, Alemayehu Amberbir, Amelia C. Crampin.

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
