## [Decision Letter · Decision Letter 0]

27 Dec 2023

PGPH-D-23-01533

Factors associated with self-reported diagnosed asthma in urban and rural Malawi: observations from a population-based study of non-communicable diseases

Dear Dr. Amoah,

Thank you for submitting your manuscript to PLOS Global Public Health. After careful consideration, we feel that it has merit but does not fully meet PLOS Global Public Health’s publication criteria as it currently stands. Therefore, we invite you to submit a revised version of the manuscript that addresses the points raised during the review process.

EDITOR: Please insert comments here and delete this placeholder text when finished. Be sure to:

Please ensure that your decision is justified on PLOS Global Public Health’s publication criteria and not, for example, on novelty or perceived impact.

We look forward to receiving your revised manuscript.

Kind regards,

Dickson Abanimi Amugsi, PhD

Academic Editor

Journal Requirements:

Additional Editor Comments (if provided):

Reviewers' comments:

Reviewer's Responses to Questions

**Comments to the Author**

1. Does this manuscript meet PLOS Global Public Health’s publication criteria? Is the manuscript technically sound, and do the data support the conclusions? The manuscript must describe methodologically and ethically rigorous research with conclusions that are appropriately drawn based on the data presented.

Reviewer #1: Partly

Reviewer #2: Yes

Reviewer #3: Yes

2. Has the statistical analysis been performed appropriately and rigorously?

Reviewer #1: No

Reviewer #2: Yes

Reviewer #3: Yes

3. Have the authors made all data underlying the findings in their manuscript fully available (please refer to the Data Availability Statement at the start of the manuscript PDF file)?

Reviewer #1: No

Reviewer #2: Yes

Reviewer #3: Yes

4. Is the manuscript presented in an intelligible fashion and written in standard English?

Reviewer #1: No

Reviewer #2: Yes

Reviewer #3: Yes

5. Review Comments to the Author

Reviewer #1: Review Reports

Title: Factors associated with self-reported diagnosed asthma in urban and rural Malawi: observations from a population-based study of non-communicable diseases.

Manuscript Number: PGPH-D-23-01533.

Review Comments

Is that published? E.g., the socio-demographic character is described in detail, however, it is extracted from the main survey.

The three triads of symptoms of Bronchial asthma were not asked in the survey/ but you have included in the figure not in the methods section/. How you supported and synthesized (a) the respondent’s response (b) the diagnosis in the survey (Who, How and Where and with what apparatus?

If it is assessment of risks, is that Descriptive or analytic? If analytic, what is the appropriate statistical approach and why you have used yours?

Lacks consistency across the document e.g., how you wrote the confidence interval, e.g., Asthma or Bronchial Asthma.

Differentiate among those associations.

Avoid redundancies.

Data quality assurance and measures taken were not addressed.

What was done for those diagnosed with ‘Asthma’?

Tense, grammar, space and hyphens and units need improvement

Lacks key words.

The tables are not self-explanatory.

The result needs further clarity and attractive ness to your reader. Try to assebmble the fragmented sub sections.

The discussion is incomplete for its contents and strength. In addition, it occupied wide large space.

Regards,

Reviewer #2: An applaudable article which meets the criteria for PLOS Global public health journal publication. The sample size justifies the attempt to answer the question posed with reported limitations. The concluded prevalence of self reported asthma in rural and urban Malawi challenges the myth/hypothesis that urbanization contributes to the rising asthma prevalence in low to middle income countries and this magnifies the relevance of the article. Although caution should be wielded before concluding such arguments, the conclusions drawn in this article are supported by data presented in the results section and analyzed with a robust statistical strategy. The cross sectional design for the primary research question posed projects an ethically and methodologically sound design with mentioned limitations/weaknesses.

An marvelous attempt at statistical analysis of the data is seen through the sub-categorization of data captured from the questionnaires and successfully shows associations and correlations between the data. The summarized version of the presentation appears in very busy and congested tables but this is understandable particularly when considering the volume of data that was processed present the statistical analysis. Additionally, the manuscript is presented in good palatable english.

Reviewer #3: The paper emerges from the analysis of the database produced in the setting of a large study with appropriately described methodology and previously published results on other subjects. The study of noncommunicable diseases in Malawi, as in LMIC in general, and the focus on differences between rural and urban populations is very pertinent and important form the public health point of view.

Table 2 is very clear-cut and provides a simple and strong message.

The paragraph of Conclusions is also good.

However, I have some suggestions that might help the authors improve the message of their analysis and results, which I share for consideration:

Is the inverse association between smoking and self-reported asthma the result of reverse causality? Or information bias due to social desirability effects? Or a mixture of both? What about moderate versus low physicial activity in rural men (lines 222).

Line 84 – the words “for which the objectives, design, populations and methodology are described elsewhere” in my opinion belong to the methods and not the background section.

In the objectives (lines 89-90) I don’t see the relevance of defining adult age as >=18. This should be clear in the methods but it does not pertain to the meaning of the objective, once the authors refer to “adults”.

Lines 112-112 – was the question nr 1 exactly as stated in the article? The grammar sounds strange (for the verb diagnosed).

The rationale to look at interactions between several variables and sex/ urban-rural on asthma outcomes is not sufficiently justified. Not in the discussion but rather in the background of statistical analysis sections.

What is the rule for the choice of 1% as significance level when attempting to account for multiple testing? The authors did much more than 5 tests, so this significance level is still high when compared to the standard 5%.

Lines 184-185 – the concept “majority of” means more than half. It is an absolute value, not a comparative one. So the idea of “compared to rural” in the final part of the sentence does not make sense.

Figure 1 would be much better and more useful with a true Venn diagram, that is, with the areas of each state and overlaps proportional to the number of people in them.

MINOR POINTS:

In the background, in the abstract and main text, the authors refer to available data in adults AS COMPARED TO CHILDREN. I argue that this point should not be seen from a relative but rather an absolute point of view. If it is insufficiently characterized in adults, it is worth the study. If it is less studied than in children but still there were enough data to deeply understand the subject, attention should be redirected to interventions.

Line 132 – the word “all” in the beginning of line 132 seems unnecessary and strange. A mistake?

6. PLOS authors have the option to publish the peer review history of their article (what does this mean?). If published, this will include your full peer review and any attached files.

**Do you want your identity to be public for this peer review?** For information about this choice, including consent withdrawal, please see our Privacy Policy.

Reviewer #1: No

Reviewer #2: **Yes: **Benson Gombe

Reviewer #3: **Yes: **Ana Azevedo

---

## [Decision Letter · Decision Letter 1]

17 Jun 2024

Factors associated with self-reported diagnosed asthma in urban and rural Malawi: observations from a population-based study of non-communicable diseases

PGPH-D-23-01533R1

Dear Dr Amoah,

We are pleased to inform you that your manuscript 'Factors associated with self-reported diagnosed asthma in urban and rural Malawi: observations from a population-based study of non-communicable diseases' has been provisionally accepted for publication in PLOS Global Public Health.

No further revisions are required by the authors

Best regards,

Dickson Abanimi Amugsi, PhD

Academic Editor

Reviewer Comments (if any, and for reference):

Reviewer's Responses to Questions

**Comments to the Author**

1. If the authors have adequately addressed your comments raised in a previous round of review and you feel that this manuscript is now acceptable for publication, you may indicate that here to bypass the “Comments to the Author” section, enter your conflict of interest statement in the “Confidential to Editor” section, and submit your "Accept" recommendation.

Reviewer #1: All comments have been addressed

Reviewer #2: All comments have been addressed

Reviewer #4: All comments have been addressed

2. Does this manuscript meet PLOS Global Public Health’s publication criteria? Is the manuscript technically sound, and do the data support the conclusions? The manuscript must describe methodologically and ethically rigorous research with conclusions that are appropriately drawn based on the data presented.

Reviewer #1: Partly

Reviewer #2: Yes

Reviewer #4: Yes

3. Has the statistical analysis been performed appropriately and rigorously?

Reviewer #1: No

Reviewer #2: I don't know

Reviewer #4: Yes

4. Have the authors made all data underlying the findings in their manuscript fully available (please refer to the Data Availability Statement at the start of the manuscript PDF file)?

Reviewer #1: Yes

Reviewer #2: Yes

Reviewer #4: Yes

5. Is the manuscript presented in an intelligible fashion and written in standard English?

Reviewer #1: Yes

Reviewer #2: Yes

Reviewer #4: Yes

6. Review Comments to the Author

Reviewer #1: Review Reports

Title:Factors associated with self-reported diagnosed asthma in urban and rural Malawi: observations from a population-based study of non-communicable diseases.

Review Comments.

A.The title is wider than the issue under study/asthma

B.The abstract still lacks clarity especially the methods section.

C. The study design should be either comparative cross sectional if the stratification as urban and rural os done before the analyasis.If its after the analysis , consider other options. This is a major and critical concern in our side.

D.The Methods section needs further elaboration for its contents e.g. the 'over lapping' occurrence of self rport and symptom as well as use of medication needs further operational definition .E.g. is that joint occurence and uae or over lapping ? I disagree with the analysis you performed as mentioned earlier E.g.ethical concerns?

E. The way you frame the result section is against most standards. E.g. associated factors before the description of characteristics of respondents .

F. Report of both CI and p value is not common and prefer one over the other based on scientific merit .

G.Language, graamer and editorial issues are also existed e.g. km without space and use of capital 'K'.

Regards,

Reviewer #2: Line 286 grammatical error ‘in contrasts to’

Reviewer #4: The study is well written, objectives clearly defined, and the same well addressed in the results and discussion.

7. PLOS authors have the option to publish the peer review history of their article (what does this mean?). If published, this will include your full peer review and any attached files.

**Do you want your identity to be public for this peer review?** For information about this choice, including consent withdrawal, please see our Privacy Policy.

Reviewer #1: No

Reviewer #2: No

Reviewer #4: No
